# A Qualitative Study on the Implementation of the National Action Plan on Antimicrobial Resistance in Singapore

**DOI:** 10.3390/antibiotics12081258

**Published:** 2023-07-31

**Authors:** Alvin Qijia Chua, Monica Verma, Sharon Yvette Angelina Villanueva, Evalyn Roxas, Li Yang Hsu, Helena Legido-Quigley

**Affiliations:** 1Saw Swee Hock School of Public Health, National University of Singapore, Singapore 117549, Singapore; monica.mh17@gmail.com (M.V.); mdchly@nus.edu.sg (L.Y.H.); ephhlq@nus.edu.sg (H.L.-Q.); 2College of Public Health, University of the Philippines Manila, 625 Pedro Gil St, Manila 1000, Philippines; smvillanueva4@up.edu.ph (S.Y.A.V.); earoxas1@up.edu.ph (E.R.); 3The George Institute for Global Health UK, Imperial College London, 80 Wood Lane White City, London W12 0BZ, UK

**Keywords:** antimicrobial resistance, implementation plan, policy development, policy analysis, One Health, Singapore

## Abstract

Antimicrobial resistance (AMR) is a global public health threat that affects humans, animals, and the environment across the One Health spectrum. Singapore launched its own National Strategic Action Plan (NSAP) on AMR in 2017 with the aim of tackling the growing threat of AMR in Singapore through coordinated approaches. However, little is known about the implementation of the NSAP. In this study, we analysed the implementation of the NSAP with guidance from an AMR governance framework. We conducted in-depth interviews with 20 participants across the One Health spectrum. The interviews were transcribed verbatim and analysed thematically. Overall, the implementation of activities against AMR was more advanced with respect to human health compared to other sectors such as (1) AMR and antimicrobial use (AMU) surveillance systems in hospitals; (2) the hospital antimicrobial stewardship (AMS) service and legislation to optimise AMU; (3) the national children and adults vaccination programme for IPC; (4) multiple avenues for education and awareness for both professionals and public; and (5) extensive research and collaboration networks with many sources of funding. Areas that were lacking presented problems including (1) an incomplete surveillance system for AMR and AMU across all sectors; (2) the need for better AMS and legislation in some sectors; (3) insufficient innovation in education for sustained behavioural modification; and (4) the need for more open research collaborations and the translation of research into policy outcomes. Improvements in these areas will enhance the overall implementation of the NSAP through a more holistic One Health approach.

## 1. Introduction

Antimicrobial resistance (AMR) is a serious global public health threat that could result in severe morbidity, death, and significant economic losses [1,2,3]. A recent assessment of the AMR burden in 204 countries showed that AMR directly and indirectly caused 1.27 million and 4.95 million deaths, respectively [4]. AMR is a natural evolutionary process that has been accelerated by human activity, spanning the One Health spectrum of human health, animal health, and the environment [1,2,5,6]. The World Health Organization (WHO) launched the Global Action Plan (GAP) on AMR in 2015 to address this multifaceted issue, setting out five strategic objectives to combat AMR and stressing the importance of the One Health approach to enabling collaboration between numerous sectors and actors including human health, animal health, environment, agriculture, finance, and well-informed consumers [5]. Countries around the world established their own national action plans (NAPs) shortly after, following the structure of the GAP.

Singapore is a highly urbanised city state in Southeast Asia with a population of 5.45 million people [7]. In Singapore’s healthcare system, 80% of primary care services are delivered through a network of private sector general practitioners (GPs), while the remaining services are covered by government-run polyclinics. Tertiary care, on the other hand, is mainly provided by public hospitals, while private hospitals cover 20% of this service. The agricultural animal health sector is much smaller than the private companion animal health sector, with three egg-layer farms and 117 fish farms as of 2019 [8]. Due to its small-scale local agricultural production level, more than 90% of food in Singapore is imported.

Singapore launched its own National Strategic Action Plan (NSAP) on AMR in November 2017 [9]. The multisectoral NSAP was jointly developed by the Ministry of Health (MOH); the Agri-Food & Veterinary Authority (AVA), which was subsequently restructured to form the Singapore Food Agency (SFA) and the Animal & Veterinary Service (AVS) under the National Parks Board (NParks) in April 2019; the National Environment Agency (NEA); and PUB, Singapore’s National Water Agency. This plan aimed to formalise existing responses, address gaps, and map future priorities across the One Health spectrum. The five core strategies listed in the NSPA are as follows: (1) surveillance and risk assessment; (2) research; (3) education; (4) prevention and control of infection; and (5) the optimisation of antimicrobial use (AMU).

In our previous studies, we found that there was good understanding of AMR as a One Health issue in Singapore. However, most of the efforts in relation to AMR control were driven by public hospitals rather than the rest of the human health sector as well as the animal and environment sectors [10,11]. At the regional level, we analysed NAPs from member states of the Association of Southeast Asian Nations (ASEAN) as a desk review and identified areas for improvement across the NAPs, including accountability, sustained engagement, equity, behavioural economics, sustainability plans and transparency, international collaboration, and the integration of the environmental sector [12]. However, we were unable to elucidate the policy processes and discussions that stakeholders engaged in during NAP development or assess the actual degree of implementation of the NAPs in practice. As of writing this article and to the best of the authors’ knowledge, no qualitative studies analysing the implementation processes of Singapore’s NSAP have been conducted. Therefore, this study aimed to address this gap using an in-depth interview (IDI) approach guided by an AMR governance framework [12].

### Conceptual Framework

In order to guide our data collection and analysis for the study, our team used the AMR governance framework published in a previous analysis of the ASEAN NAPs [12]. The framework was developed by restructuring a prior published framework developed by Anderson et al. [13], featuring five domains on policy design, implementation tools, monitoring and evaluation, and sustainability, with One Health engagement situated in the centre (Figure 1).

## 2. Materials and Methods

### 2.1. Study Population and Data Collection

This study was conducted from November 2020 to October 2021. Purposive sampling was used to recruit relevant One Health stakeholders for AMR. Potential participants were selected from organisations relevant to the development and execution of the NSAP through publicly available organisation charts and the Singaporean government’s public directories. In addition, we also conducted snowball recruitment based on participant referrals and recruitment through the research team’s personal contacts. Potential participants were contacted via email, thereby providing the study information and request for participation. Up to two reminder emails were sent, and if there was no response, the invitation was considered rejected.

A total of 40 potential participants were contacted. Twenty agreed to participate, representing stakeholders from government agencies, healthcare institutions, academia, and private industry (Table 1). The human health sector presented the largest number of participants. There was difficulty recruiting participants from the animal health and environment sectors, which was likely due to the perceived smaller role that they play in AMR. Fifteen of those contacted did not respond, and the remining five declined participation due to work commitments related to the COVID-19 pandemic.

In view of the COVID-19 pandemic restrictions, interviews were conducted virtually over Zoom in English, with only the researchers and interviewee present. Interviews were conducted by up to two researchers (A.Q.C. and H.L.-Q.), both of whom were trained in qualitative research. The interviewers were a Research Associate and an Associate Professor. There had been no prior engagement nor were there any established relationships between the researchers and the participants. Each interview, which lasted 60 min (on average), was audio-recorded. A semi-structured question guide was used to explore our research aim in relation to the participants’ fields of expertise and interests (Appendix A). The question guide was developed after a literature review and the achievement of a consensus among the researchers [9,10,11,12,13]. There were no follow-up interviews, and participants were not remunerated for the interview.

### 2.2. Analysis

Interviews were transcribed verbatim, and the QSR NVivo software (Release 1.5.1) was used for data management. Two researchers (A.Q.C. and M.V.) independently read and coded the transcripts. We adopted an interpretative approach to analysis, focusing on participants’ perceptions of the discussion topic. The initial coding process was performed deductively, using the AMR governance framework as a guide [12]. Themes were allowed to emerge inductively if the codes did not fit into the framework. We drew on techniques from the grounded theory such as line by line analysis and the use of the constant comparative method to code in an iterative manner [14,15]. Discussions between the two researchers were organised to resolve any disagreements and finalise the themes. Thematic saturation was established when, though discussion, the research team agreed that there were no additional insights identified from the data. A member check was conducted at the final stage of the manuscript’s preparation to ensure that the participants’ perspectives were represented accurately and to validate our interpretation of the data. Findings were reported according to the COREQ checklist (Appendix A).

### 2.3. Ethical Considerations

This study received ethical approval from the National University of Singapore, Saw Swee Hock School of Public Health Department Ethics Review Committee (SSHSPH-012). An information sheet detailing the objectives and methods of the study, as well as the confidentiality and anonymity of the participants’ responses, was shared with potential participants during recruitment. Prior to the start of each interview, the interviewers obtained verbal consent from the participants to audio-record the interview session and include their anonymous quotes in research outputs. Participants had the option to reject any of these as well as any questions posed to them during the interview itself. All records associated with the participants were de-identified to maintain confidentiality.

## 3. Results

We present our findings based on the following eight main themes: surveillance, AMU, infection prevention and control (IPC), education, research and innovation, international collaboration, monitoring and evaluation, and sustainability. These are based on relevant domains in the AMR governance framework. Selected quotes from each theme are presented in Table 2, Table 3, Table 4, Table 5, Table 6, Table 7, Table 8 and Table 9 for each of the themes presented below.

### 3.1. Surveillance

#### 3.1.1. AMR and AMU Surveillance

In the human health sector, several participants stated that there was routine reporting on the incidence of resistance isolates in both public and private hospitals, which served as an alert mechanism for the early detection of emerging resistance. Many participants also described AMU, healthcare-associated infection (HAI), and IPC measures surveillance including hand hygiene compliance surveillance and the appropriate cohorting of patients in hospitals. Surveillance of hospital effluents was only performed as part of research, with sampling for neither drug-resistant organisms nor antimicrobial residues in effluents performed routinely.

Regarding primary care, one participant stated that there was regular culturing of urine samples as part of the workup for urinary tract infections in polyclinics. This was, however, not routine for other infections such as acute respiratory infections (ARIs). The participant also mentioned the need for better community detection of patients with MDROs. AMU surveillance in the primary and long-term care setting was described to be more difficult. One reason shared commonly was the large number of private GP clinics.

In the animal health sector, there was ongoing AMR surveillance of non-food animals, albeit opportunistically. Some participants stated that stray animals undergo a ‘Track, neuter, release’ programme, where they are swabbed for selected resistant microorganisms. As for wildlife and companion animals, samples were collected during field work and as part of disease workups by veterinarians, respectively. AMU surveillance with respect to animals was lacking. Several participants highlighted that there was no clear understanding of the type and quantity of antimicrobials used, and a proxy based on antimicrobial sales data from wholesalers was used instead. However, one participant stated that there were plans to improve surveillance on this matter.

As for food products, a few participants mentioned regular food testing, including with respect to imported frozen and chilled meats for selected microorganisms and antimicrobial residues, although the sampling methodology employed was unclear. In addition, the focus in this regard was on food safety rather than AMR. One participant stated that surveillance should be integrated with other sectors instead of engaging in aggressive sampling within the food-animal sector.

Regarding the environmental sector, specifically in water bodies and supply systems, a few participants stated that AMR and antimicrobial surveillance was largely research-based or was part of surveillance of environmental contamination.

Several participants mentioned that a national system across various sectors was being developed. However, there were concerns regarding this national-level surveillance programme. For example, one participant stated that sectors might have their own areas of focus and concern.

#### 3.1.2. Laboratory Capacity

Adequate laboratory capacity and ability supported by regular external quality assessments is essential for surveillance efforts. Regarding the human health sector, a few participants discussed differences between public and private hospital settings, highlighting the lack of technical expertise in the latter.

Concerning the animal health sector, some participants mentioned that the testing of companion animals and food samples was conducted by the NParks and SFA laboratories, respectively. Private laboratories were also engaged in these activities. Some larger food animal farms such as fish or shrimp farms established in-house laboratories to ensure good water quality.

A few participants highlighted that the environmental sector did not have sufficient resources to handle surveillance work and thus engaged external parties, including academic research groups, for support. However, they mentioned the presence of more advanced technology to support AMR surveillance.

**Table 2 antibiotics-12-01258-t002:** Subthemes under the theme of surveillance and quotes selected from among them.

Subtheme	Sector	Representative Quotes
AMR surveillance	Human health	Hospital “What’s shared within the infectious diseases community is this annual survey… nosocomial multi-drug resistant Gram-negative, ESKAPE pathogens, MRSA… There’s a timeline and trend on the bacteria of interest.”—IDI05, Human Health Primary care “…for children with UTI suspicion, we would routinely order urine culture while we initiate antibiotics… acute respiratory infections are a little bit tricky because we don’t routinely culture sputum.”—IDI06, Human Health “We need new policies to handle early MDRO detection these patients, rationalise what to do when these patients are in the community… identify if they will be as deadly and infectious as COVID-19, or just be a latent presence in the body.”—IDI06, Human Health
	Animal health	“I feel that the animal people don’t understand that ultimately, the concern is how AMR affects humans… they tend to go on their own rather than ask us what our priority is. They were looking for ESBL in poultry, but ESBL is already widespread in hospitals, plus the main driver is hospital spread and not animals… what we want to monitor seems quite different.”—IDI08, Human Health Food animals/products “…it’s part of a broader surveillance system, not just for AMR. The whole purpose of food surveillance is for food safety… since we’ve got those isolates, we’ll look at the antimicrobial profile.”—IDI17, Human Health “We are not privy to the way they test… For example, one whole big crate of chicken… they will just pick one or two to sample. We are not sure whether that’s good enough as a sampling point.”—IDI02, Human Health “Singapore is proud of testing a million samples for specific pathogens every year from incoming food… It’s very unlikely that you can assess the level of important AMR pathogens this way. In general, it doesn’t help to just do testing… you should test in a strategic way so that you end up with relevant estimates of the level of AMR in pathogens as well as in other relevant bacterial species… and importantly, you should test with the same methodology in the different sectors so that you can compare the prevalence of specific AMR in bacteria in food and humans, which in the end can give you indications of the root of the problem.”—IDI19, Animal Health
	One Health	National AMR surveillance “…this national surveillance system is an area that was highlighted before. We recognise the advantages that come with being able to look at data more holistically. It’s something that we’re working on, but it will take significant resources and capacity to set up.”—IDI14, Animal Health
AMU surveillance	Human health	Primary care “We don’t have a mechanism to collect antimicrobial use data nationally… the market share for GPs is about 80% whereas polyclinic is 20%. Each GP is their own private entity… we don’t have IT systems that allow us to pull out that data.”—IDI12, Human Health
	Animal health	“We’re supposed to track the amount of antimicrobials that each animal industry used. But because we don’t go down to all users to check how much antimicrobials they use, we’re tracking it from wholesalers who sell them… It’s on a voluntary basis.”—IDI10, Animal Health “We intend to introduce some requirements on recordkeeping, for example on sale, distribution, import, and manufacture…”—IDI14, Animal Health
Lab capacity	Human health	Private hospital “…there’s a huge gap when it comes to microbiology and technical expertise. What we have is very rudimentary… We don’t even have a microbiologist right now across the entire [organisation]. I’m willing to ask management to pay top dollar to attract somebody from the public sector…”—IDI03, Human Health
	Environment	“AMR is not a new issue. Even though it’s not a concern, we want to know what is in our water. With advanced technology and capability to do metagenomics, high throughput sequencing, you can now establish that even faster.”—IDI01, Environment

MRSA = methicillin-resistant *Staphylococcus aureus*; UTI = urinary tract infection; MDRO = multidrug-resistant organism; ESBL = extended-spectrum beta-lactamase; AMR = antimicrobial resistance; GP = general practitioner.

### 3.2. Optimising AMU

Most participants felt that this domain was the most developed in the human health sector and could be improved in the animal health sector.

#### 3.2.1. Antimicrobial Stewardship

The majority of the participants mainly discussed antimicrobial stewardship (AMS) when asked about optimising AMU. They defined AMS as multifaceted programmes that involved reviewing AMU with regard to patients, formulary restrictions, AMR and AMU surveillance, and education to relevant stakeholders. More recently, a few participants reported that antimicrobial stewardship programmes (ASPs) were venturing into services including therapeutic drug monitoring for antimicrobials to optimise dosing for individual patients and rapid diagnostics for the early detection of multidrug-resistant organisms (MDROs). One participant stated that rapid diagnostics complemented ASPs and improved patient outcomes. However, there were implementation challenges due to logistical and acceptance issues. In private hospitals, ASPs were described to be non-mandatory, although there were efforts to establish them. One participant mentioned that their ASP was basic, with little support from hospital administrators, although they mentioned that it had been improving over the years. There were other challenges to implementing ASPs in private hospitals compared to their public counterparts. For example, because most private patients were from overseas, there were more MDRO infections. Physicians also had limited access to patients’ clinical details either because they are from overseas or because of privacy concerns.

In the primary care setting, ASPs were described to be lacking and present only as educational webinars on appropriate AMU with respect to common infections for primary care physicians. One participant attributed this to a lack of a good model for AMS in primary care and further stated that AMS was better managed in the polyclinic setting compared to private GPs.

Treatment guidelines development was highlighted as another way to optimise AMU. Some guidelines that were mentioned included the national ARI guidelines and surgical prophylaxis guidelines [16]. A few participants felt that there were insufficient guidelines in the primary care setting and that existing ones were outdated. One participant stated that improper diagnoses constituted a major problem even when AMU guidelines were in place. 

There were no ASPs in the animal health sector, but ASPs were highlighted in the Prudent Use of Antimicrobials Guidelines to promote judicious AMU among veterinarians [17]. Some participants provided reasons why these programmes were difficult to establish, including the large variety of animal species involved and the large number of small private veterinary clinics in Singapore.

#### 3.2.2. Regulation of Medicines

Developing and enforcing regulations helped ensure appropriate AMU. Many participants stated that various governmental authorities monitored and enforced legislations well with respect to ensuring appropriate AMU. They highlighted that the human health sector imposed tight regulations on access to antimicrobials, which were prescription-only medications, i.e., they were only purchasable with a valid physician’s prescription. Despite this, there was concern that physicians themselves are not regulated and that any physician can prescribe any antimicrobial available on the market. A few participants stated that it was challenging to develop a legislation that would satisfy all stakeholders. For example, a participant talked about dispensing rights, which would help improve appropriate AMU against a backdrop of revenue generation.

The regulations in the animal health sector were described to be inferior to those of the human health sector. AMU for growth promotion is not allowed, and all veterinary clinics and pet shops that sold live animals must be licensed. Some participants stated that licensing is the main way of regulating AMU, as there is no legislation preventing antimicrobial sales; there were instances where pet owners purchased medications from overseas or online sources. They also stated that there were efforts to improve legislation for AMU with respect to animals, although the process was complicated because many products formulated for humans were used by animal patients in veterinary clinics.

The agricultural sector was described to be less problematic, as high-quantity users on farms did not use human products due to cost issues. However, there was incomplete information on AMU due to the voluntary nature of the licensing and data submission system. In order to establish better AMU oversight, plans to develop a mandatory licensing system for wholesalers of antimicrobials for agricultural use are under development.

Regulations on antimicrobials in the environment were rarely discussed. Some participants mentioned regulations on appropriate antimicrobial disposal in their households overseas but not in Singapore.

**Table 3 antibiotics-12-01258-t003:** Subthemes under the theme of optimising antimicrobial use and quotes selected from among them.

Subtheme	Sector	Representative Quotes
Antimicrobial stewardship programmes	Human health	Public hospital “The process was hampered by many things… machines were not working, processes delayed. The definition of rapid diagnostic is for them to be rapid, but you find that it takes a whole day… There’s a cost issue… I also gather that not every infectious diseases physician agrees with this concept….”—IDI03, Human Health Private hospital “ASP is not mandatory in private hospitals… those who believe that it is important to have ASPs are facing challenges in setting up.”—IDI17, Human Health “…there was a very rudimentary service… a very limited scope, perhaps just a vancomycin audit… there was very little support and training amongst pharmacists as well… that was the setup that we were faced with.”—IDI03, Human Health Primary care “…there’s oversight in the polyclinics on how much antibiotic is prescribed for a labelled diagnosis… But in the private GP setting, there is no such mechanism, and people may not be aware that they are prescribing more than their peers, other GPs.”—IDI12, Human Health
	Animal health	“The difficulty for ASP in animals is that nobody is sure what it is supposed to look like… there’s so many species involved, so many different settings.”—IDI17, Human Health “The vet industry is made up of many small private clinics. We don’t have any public hospitals unlike the human side… We can’t dictate what they do and what they use. It is up to individual clinics to decide whether they want to implement any programme…”—IDI14, Animal Health
Treatment guidelines	Human health	“The diagnosis problem came because of protocols for antibiotics utilisation in the hospital. For example, if a patient has healthcare-associated pneumonia, you can use Pip-Tazo and vancomycin… In the end, anyone who gets admitted, especially during on-call, are diagnosed with healthcare-associated pneumonia, even though they have a psoas abscess… This is a problem and I’m not sure whether anyone is addressing it.”—IDI07, Human Health
Medicines regulation	Human health	“…I heard that primary care doctors in Australia can’t even prescribe fluoroquinolones without some stewardship oversight. Over here, every practitioner can give any antibiotics that’s in the market… last week, one patient was given five days of home-administered IV meropenem by a GP, prior to hospitalisation.”—IDI12, Human Health “At one time, the ministry of health was contemplating to have medicines dispensed at community pharmacies instead of GPs. In Asian countries, GPs prescribe as well as dispense medicines… that’s where they have revenue… from medicines sales to patients… In the West, they have sole dispensing rights in community pharmacies.”—IDI06, Human Health
	Animal health	“We are controlling through vets that we license… assuming that you get your antimicrobials from a licensed vet with a prescription. There is no legislation that we can use to stop pet shops from selling antibiotics. But we license pet shops so we can control them …”—IDI10, Animal Health
	Environment	“I think overseas, they have rules and guidance… they are not allowed to discard their antibiotics down their toilet bowls. In Singapore, we don’t have that kind of guidance.”—IDI13, Environment “I don’t think we have regulations here in Singapore. For coastal waters, I know there’s a certain limit for faecal coliforms, like *E. coli*, *enterococcus*, but in terms of antibiotics, I don’t think that there are standards in place.”—IDI20, Environment

ASP = antimicrobial stewardship programme; GP = general practitioner; IV = intravenous.

### 3.3. Infection Prevention and Control

Regarding the human health sector, some participants stated that the MOH has published various guidelines and standards on IPC. They also highlighted the National Immunisation Programme, which provides schedules for both adults and children for various vaccines. One participant related that the government encouraged vaccination through subsidies.

Many participants mentioned that hospitals have implemented IPC strategies such as screening patients for MDRO colonisation, isolation, and cohorting policies as well as hand hygiene and sanitation practices. They also highlighted surveillance measures such as swabbing hospital environments for MDROs and engineering measures such as appropriate ventilation and bed spacing. Some participants mentioned that hospital effluents were treated before being released into sewers.

Regarding primary care, one participant related their IPC practices for discharged patients with MDROs, highlighting that there were rooms reserved for these patients to have their procedures performed, but this came with challenges as it was difficult to determine if these patients still carry the MDROs and whether the microorganisms were colonisers or infectious.

Concerning the animal health sector, some participants discussed the development of guidelines, including the Singapore Vaccination Guidelines for Dogs and Cats and the Pet Husbandry Guidelines. There were also biosecurity programmes including vaccinations and facility management as well as stakeholder licensing to ensure appropriate IPC in the agricultural setting. A few participants also discussed IPC measures for animal importation. Food animals that are imported live must be certified as healthy before their entry into Singapore. In addition, assessments of the animal sources were undertaken to ensure that appropriate biosecurity measures were being employed. Companion animals undergo similar health checks and certifications before they are allowed into Singapore. One participant stated that the conducted screenings were aligned with international standards from the OIE but that only certain diseases were screened for.

Regarding the environmental sector, most discussion concerned IPC measures in drainage and sewage systems. Some participants stated that having separate drainage systems for storm water and sewage helped prevent contamination, although this measure did not target AMR specifically. One participant highlighted that Singapore’s water treatment plants employ advanced treatment technologies such as membrane-based processes, activated carbon, and ozonation. In addition, chlorine is added for disinfection to prevent microbial growth during distribution in the pipes. Risk assessments are performed to ensure appropriate personal protective equipment use by staff working in the treatment plants to protect them from exposure to infections and other contaminants in the pre-treatment water.

**Table 4 antibiotics-12-01258-t004:** Subthemes under the theme of infection prevention and control and quotes selected from among them.

Subtheme	Sector	Representative Quotes
Infection prevention and control	Human health	Hospital “Effluents from high dependency and isolation ward… there’s always a last tank where they will have to dose sodium hypochlorite, before discharging to our sewers.”—IDI13, Environment Community “Uptake for certain vaccinations is really good… like for children. But there’s room for improvement for pneumococcal and flu vaccinations in the 65-and-above age group. MOH has given some subsidies, rolled out last November, to support those activities.”—IDI11, Human Health Primary care “When these patients are discharged from hospitals, they come to our clinic for wound dressings. We would have identified a room for them to do the procedures. There is power cleaning before the room can be used for another patient.”—IDI06, Human Health “When these patients are discharged from the hospital, we are not aware if they still have these MDROs. There are talks about screening for example, MRSA. But doing swabs has its difficulties. We have to send the samples for analysis, when the patient is already in our clinic. Also, it is unknown if the strain is infectious…”—IDI06, Human Health
	Animal health	“We license stakeholders who work with animals in a group setting, for example dog breeders. There are controls in place to ensure adequate biosecurity hygiene, which will tie in with IPC.”—IDI14, Animal Health “When companion animals are imported into Singapore, there are control measures in place for certain diseases, for example rabies… The quarantine duration depends on the country risk tier… We are not testing animals to look for AMR…”—IDI10, Animal Health
	Environment	“…we have a separate drainage system for storm water and sewage. We don’t have wastewater influence in our freshwater body… If you have a combined sewer… it can overflow when there is a heavy rainfall. And then that, tainted by sewage runoff will enter catchment which subsequently goes to water bodies for various sources… for recreational or for treatment into drinking water.”—IDI01, Environment

MOH = ministry of health; MDRO = multidrug-resistant organism; MRSA = methicillin-resistant *Staphylococcus aureus*; IPC = infection prevention and control.

### 3.4. Education

#### 3.4.1. Education for Professionals

Several participants mentioned certifications and programmes for professionals to ensure adequate knowledge and understanding to effect strategies for tackling AMR. These included AMR topics in the school curricula of healthcare and animal health professionals. Some participants highlighted the importance of education on proper diagnoses and appropriate antimicrobial selection and dosing.

Continuing education (CE) programmes for physicians were discussed. Various AMR topics, taught through seminars in both hospital and primary care settings, were mentioned, including appropriate diagnostic tests, treatment guidelines, and vaccination protocols. One participant mentioned that newsletters from medical professional organisations constituted another educational platform. AMS CE programmes and awareness campaigns were discussed frequently. These campaigns, covering topics including the concept of AMS, the role of the AMS team, and current prescribing problems, were held at venues including roadshows and organisation townhall sessions. One participant mentioned that hospitals with less-established ASPs sent their pharmacists for training at hospitals with established programmes or overseas courses.

Efforts in the animal health sector were less extensive than those in the human health sector, mainly consisting of guidelines on prudent AMU and workshops on biosecurity measures, as described earlier.

Overall, many participants voiced concerns about the effectiveness of these CE programmes. Some thought that education was only provided at the surface level and that behavioural change was difficult to instil. This was accompanied by the fact that despite a lack of results, the mode of delivery has not changed over the years. There was also a comment that these programmes may not be targeting the right audience because perhaps only those who were interested signed up for a given programme.

#### 3.4.2. Public Education and Awareness

Public education and awareness campaigns were discussed extensively by all participants. Many participants discussed public outreach on various media platforms including exhibiting campaign collaterals at bus stops, metro stations, various forms of public transport, newspapers, television programmes, and social media platforms, many of which were launched parallel to the World Antimicrobial Awareness Week in November each year. Some of them mentioned that these media channels provided a platform for physicians to introduce AMR to their patients during consultations, which, ultimately, might influence prescribing or vaccination practices. For school children, basic AMR knowledge was included in their primary or secondary school curricula. There were also awareness campaigns organised at public libraries. One participant suggested that teaching school children may help educate their parents as well.

Concerning the animal health sector, there were outreach programmes targeting specific populations. For example, one participant stated that during the annual Pets’ Day Out event, pet owners were educated on preventive medicine and early treatment.

The participants highlighted challenges related to communication with the public to increase awareness on AMR. Some felt that existing mass-messaging efforts were short-term and might not be effective and that changing the public mindset required more than just education – the campaigns should factor in the supply- and demand-side behaviour patterns and address the social determinants that drive such behaviour regarding AMU. In addition, a few participants stated that public communications could be clearer and better targeted toward specific populations.

**Table 5 antibiotics-12-01258-t005:** Subthemes under the theme of education and quotes selected from among them.

Subtheme	Sector	Representative Quotes
Education for professionals	Human health	“…to educate the healthcare sector… one main problem with antibiotic use is not whether you choose the right antibiotic… in the first place, you must diagnose infection correctly.”—IDI08, Human Health “Different medical groups have different ways of communicating to registered providers… for example, an article in their e-newsletter written by another physician… that’s really targeted information that physicians would want to hear… this might drive some behaviour change.”—IDI11, Human Health “This year, we are discussing with clinicians through a CE programme and regular townhall sessions the fact that antibiotic prophylaxis should be more judicious… there are many clinicians who keep giving prophylaxis beyond the recommended duration.”—IDI03, Human Health “Education has been beaten to death for many years now, so why is it not working? Either the educators are not doing their job properly or people who are learning are not really learning. You can’t do the same thing again and again and expect a new result. But unfortunately, that’s what I see. We just keep doing the same thing…”—IDI07, Human Health “Sometimes we don’t know whether we are preaching to the choir, where the people who are already interested and doing what they should do, are signing up…”—IDI12, Human Health
Public education and awareness	Human health	“…we can remind our patients with posters and media release regarding this topic so that they will know that it is something they do not need. It will be easier for us to persuade them to not receive antibiotics at all.”—IDI06, Human Health “… on education to GPs or doctors in general, we supported through development of a resource… an A5-size standee which talks about the role of antibiotics, side effects of antibiotics and why you shouldn’t take antibiotics for viral infections. We have disseminated it to all GPs and health healthcare institutions… I think sometimes the doctors feel pressurised to prescribe antibiotics to the patients, so we wanted the simple infographics to help them explain…”—IDI15, Human Health “…they can go home and tell their parents that these are the things they learnt… the parents will then learn how to look after them better…”—IDI06, Human Health
	Animal health	“In most of our events, like the Pets’ Day Out for any animal or pet-related events, AMR now has a dedicated space for us to educate pet owners on this issue.”—IDI09, Animal Health “We try to bring across to pet owners the importance of preventative health… making sure that they have their pets vaccinated, and that they are aware of common issues that pets face… Sometimes skin issues start with a small patch that the dog licks or scratches at… There’s a need to have these worked up and managed early, before it develops a secondary bacterial infection… this will negate the need to use antimicrobials.”—IDI10, Animal Health

CE = continuing education; GPs = general practitioners, AMR = antimicrobial resistance.

### 3.5. Research and Innovation

The participants provided examples of AMR research collaborations. One initiative mentioned by many participants was the One Health Antimicrobial Resistance Research Programme (OHARP), which encouraged cross-sectoral collaboration. OHARP awarded funding to projects that spanned at least two sectors, for example, AMR research on pets and their owners. Many participants also discussed other collaborations, including national and international research groups, and collaboration between hospitals, governmental agencies, private organisations, and academic institutions. One participant stated that collaborations usually remain restricted to research rather than progressing to the operational stage due to differences in each stakeholder’s key performance indicators (KPIs). A few participants voiced that, in their experience, these collaborations are not truly open due to concerns regarding data theft.

Various channels of dedicated budgets for AMR research were discussed, including that provided by OHARP. All agencies in AMRWG contributed to this grant. However, some participants stated that it took a few years to establish and that the amount was small compared to other national grants. Other sources included international funding from the WHO, national research grants, and individual agency grants. One participant highlighted that it was challenging to obtain funding for their research work as their topic was not the funding agencies’ focus.

A few participants highlighted various types of R&D innovations and AMR research that were made and conducted, respectively. In the area of R&D, the participants highlighted new rapid detection technologies for resistant microorganisms as well as new treatment options. One participant discussed using phage therapy for bacterial infections along with some regulatory issues during the therapy’s approval process.

Other types of AMR research were also discussed. Some examples that were shared included understanding the mechanisms of resistance of Gram-negative bacilli and decolonisation from the human body, the whole-genome sequencing of MDROs and its applications (such as understanding AMR transmission across sectors), and the role of hospital environments as a source and reservoir of MDROs. There was also research undertaken to understand the knowledge, attitudes, and perceptions (KAP) of various stakeholders with respect to AMR. Some of this research was conducted over social media platforms and/or using the National Population Health Survey.

A few participants stated that the goal of research is to fill knowledge gaps and influence policies or work processes. One success highlighted was the conversion of research on the combination testing of antimicrobial treatment of MDROs into clinical services. Another innovation mentioned was the use of AI to screen patient cases worthy of AMS audits as a replacement for manual review performed by humans. The participant hoped that this innovation, which is currently still at the research stage, could be expanded to the national level.

There were also areas that required improvement. One participant highlighted that despite KAP studies, there were no policy outcomes that resulted from research. Overall, there was a sense that more research on AMR is required, especially in the animal and food sectors and in the primary care setting.

**Table 6 antibiotics-12-01258-t006:** Subthemes under the theme of research and innovation and quotes selected from among them.

Subtheme	Sector	Representative Quotes
AMR research	Human health	“[Organisation] doesn’t award translational research… As a clinician, translational research is something that gives direct benefit to patients at the end of your study. That’s not what [organisation] looks at… for them, it is more about finding new molecules, from a viral or bacterial genetic marker to a product. If you talk about HAI, ‘oh, it’s not chic enough’…”—IDI07, Human Health “We are always understanding people’s perceptions, opinions, and problems, which are very important. But after understanding, it should be followed up with strong policy measures, but there’s none.”—IDI07, Human Health
	Environment	“For AMR, it will be research… to operationalise, we will have to do it from our end… We can get [university] to support our operation, but their students cannot graduate with papers… we are mindful that they have KPIs at the research end. We try to have projects where they can deliver their KPIs and we get what we want, so it’s a win-win situation.”—IDI13, Environment
	One Health	“There’s this nervousness that if we collaborate too closely, they will take our data, while in reality you can only get a true picture through the comparison of data between sectors. I think the collaboration was not totally open and therefore not efficient between the sectors.”—IDI19, Animal Health “…it took three years to run the grant call… We could not find a specific platform to roll out this research. In the end, it was agreed at the One Health workgroup that we will roll this out ourselves… each agency gave different proportions of money to this fund.”—IDI13, Environment
AMR innovation	Human health	“New antibiotics are not the way to go. We need to explore something that is easy to purify and procure, and must be pathogen specific… it should kill bacteria, but keeps the microbiome intact… I’m attempting phage therapy but our problem here is that regulators expect phage to go through the usual antibiotic approval framework… You can’t, because when you bottle up a phage, it’s going to change over time as it is alive…”—IDI02, Human Health

AMR = antimicrobial resistance; HAI = healthcare-associated infection; KPIs = key performance indicators.

### 3.6. International Collaboration

The participants provided some examples of international collaborations and partnerships. At the political level, a few participants highlighted participation in the One Health Global Leaders Group on AMR, which was set up to strengthen global political momentum and leadership regarding AMR. Regionally, there was collaboration on AMR work between ASEAN countries.

Individually, agencies and stakeholders from various sectors across the One Health spectrum have collaborated with international organisations on AMR-related work, including the WHO, the US CDC, the FAO, the Global Water Research Coalition, and the US EPA. There were also partnerships on a smaller scale, for example, international hospital exchanges to learn about best practices in AMS work and the organisation of regional training courses on AMS. One participant mentioned working with private companies such as bioMérieux to develop rapid diagnostics for human health.

**Table 7 antibiotics-12-01258-t007:** Subthemes under the theme of international collaboration and quotes selected from among them.

Subtheme	Sector	Representative Quotes
International collaboration	One Health	“When the Philippines was the ASEAN chair, they got leaders to issue a declaration against AMR in their countries. There was a commitment… we worked together on drawing up a framework that each country could follow as they do their NAPs.”—IDI04, Human Health

ASEAN = Association of Southeast Asian nations; AMR = antimicrobial resistance; NAPs = national action plans.

### 3.7. Monitoring and Evaluation

#### 3.7.1. Effectiveness

Many participants stated that although there were no targets within the NSAP, assessments of the implemented initiatives were conducted. Regarding the human health sector, one aspect mentioned frequently was the effectiveness of ASPs. This was evaluated through a review of the rates of intervention acceptance, appropriate prescribing, and antimicrobial consumption. Many participants highlighted that there was an overall improvement in intervention acceptance and appropriate prescribing rates. One participant highlighted that an occasional lack of improvement could be due to suggestions to carry out bolder interventions, such as stopping a course of antibiotics instead of de-escalating to an antibiotic with a narrower spectrum, which were harder to accept by the primary care team. However, despite the ASPs, the participants reported no change in overall antibiotic consumption rates.

Another area frequently mentioned was the assessment of public awareness campaigns. Surveys were administered to evaluate the participants’ understanding and perception of campaign messages. One participant mentioned that the level of understanding on AMR was not optimal because about 50% of the survey participants did not know that antibiotics are ineffective against the flu.

Some participants voiced concerns that parameters related to awareness and education were difficult to measure, including public acceptance of not using antimicrobials and behavioural changes. It was revealed that one way of measuring the effectiveness of public awareness campaigns, specifically immunisation campaigns, is to review the post-campaign immunisation rates provided by the National Immunisation Registry (NIR).

Concerning the animal health sector, some participants highlighted that the effectiveness of implementation plans was only assessed in the form of antimicrobial sales data from wholesalers. At the international level, Singapore participated in the Tripartite AMR Country Self-Assessment Survey (TrACSS), which highlighted the progress made on the implementation of NAPs.

#### 3.7.2. Reporting and Dissemination

The effectiveness of implementation plans was reported at various levels, including institutional, ministry, One Health, and international levels.

At the One Health level, most participants described the reporting of each agency’s implementation plans during regular AMRWG meetings.

Beyond AMRWG, each of the representatives reported to their own agencies and senior managements. Within the MOH, KPIs were also reported to specific committees. For example, hospital data on (1) AMU, AMR, and ASP indicators were reported to the National Antimicrobial Resistance Control Committee (NARCC), while (2) HAI and IPC-related data were submitted to the National Infection Prevention and Control Committee (NIPC). These were also reported to hospitals’ senior management staff regularly. National vaccination rates were also reported to the MOH. This was executed through manual reports provided by every healthcare provider to the NIR. A few participants noted that such data might not be accurate, but such accuracy has improved owing to subsidy schemes employed by the MOH in recent years.

At the international level, Singapore had been contributing data to international platforms, including the WHO Global Antimicrobial Resistance and Use Surveillance System (GLASS), the OIE World Animal Health Information System (WA-HIS), and TrACSS.

With regard to dissemination, many participants mentioned the annual report by the AMRWG, ‘One Health Report on Antimicrobial Utilisation and Resistance’, which detailed surveillance activities for AMR and AMU across the human, animal, food, and environment sectors [8]. This surveillance report is available online publicly. Some participants also mentioned that a report on the progress of NSAP implementation plans is under development.

#### 3.7.3. Feedback Mechanisms

Several participants stated that during AMRWG meeting updates, stakeholders discussed why certain targets were not met in time and then updated plans to achieve these targets.

At the national level, some participants described that the MOH followed up on submitted reports on hospital AMR and AMU surveillance. These reports were also used to benchmark performances among different hospitals, although, officially, objective comparison was difficult because of the different practices and patient populations. In addition, benchmarking exercises triggered further investigation by those who performed poorly.

Locally, at each institution, hospital ASP KPIs were reviewed and fed back to relevant stakeholders. For example, cases of inappropriate AMU were reported along with the provision of relevant recommendations. These recommendations were provided at the individual or departmental levels and at a larger scale through CE programmes or town hall sessions. ASP KPIs were also linked to monetary benefits, for example, institution bonus metrics for encouraging appropriate antimicrobial prescribing. Unfortunately, the above feedback mechanisms were more difficult to implement in private settings. At the primary care level, a similar review of AMU was conducted to examine the gaps in prescription patterns.

Other feedback mechanisms in the community setting were mentioned. On the education front, a few participants described the usage of data collected from pre- and post-campaign surveys to guide future campaigns, including with respect to the selection of primary messages and target audiences.

In the environmental sector, surveillance reports were used to guide the selection of targets for detection prospectively.

**Table 8 antibiotics-12-01258-t008:** Subthemes under the theme of monitoring and evaluation and quotes selected from among them.

Subtheme	Sector	Representative Quotes
Effectiveness	Human health	Hospital “Initially, about 60% of prescribing was inappropriate… with pharmacist interventions, we managed to get optimisation of use to approximately 80%. Acceptance rates initially around 60% have now come up to about 80%…”—IDI03, Human Health “…attitudes are changing, acceptance rates are improving. Sometimes even if it doesn’t improve, it’s because we are picking challenging cases to intervene, making harder interventions for people to accept… we say stop antibiotics instead of de-escalating…”—IDI12, Human Health Community “For AMR, about 50% of the people do not have the right knowledge. Only half know that antibiotics do not work on flu. We still have work to do in public education.”—IDI15, Human Health “We are always discussing about how to get the best reach that actually impacts people to take a different behaviour… that’s really hard to measure… will it actually result in behaviour change?”—IDI11, Human Health “…one of the things that we look at is whether there is a positive change in data that tracks behaviour change. For example, after a vaccination intervention or a subsidy roll out for pneumococcal and flu vaccination, we would look for increase in vaccinations in the following months.”—IDI11, Human Health
	Animal health	“The only thing that we’re tracking right now is sales data from wholesalers. We don’t have a programme that measures how effective it is… we’re not tracking the rate at which AMR is developing… this is something which is in the works.”—IDI10, Animal Health
	One Health	“The tripartite organisations FAO, OIE and WHO send out a survey every year… some progress was made when it comes to increasing awareness and pushing out things to stakeholders. On the vet medicine side, there’s a big piece which involves development of legislation to regulate vet medicine. Because it requires changing of acts, it has a slightly longer timeline… once that is in place, we have clarity over the products that we’ll be able to track… how much is sold.”—IDI10, Animal Health
Reporting and dissemination	Human health	Community “…there are some discrepancies… But with subsidies that were rolled out last November, there’s a stronger incentive now for subsidised clinics to input data into NIR, in order to get their return on subsidies.”—IDI11, Human Health
	One Health	“The AMRCO will compile a report for the taskforce. We haven’t done that kind of report yet, but the intention is that at the end of five years we will do a report to evaluate how successful we have been…”—IDI04, Human Health “Updates are given on activities that they’re undertaking. Whatever their agency is responsible for, they will report back to this group on progress, timelines of grants…”—IDI11, Human Health
Feedback mechanisms	Human health	Public hospital “The ministry is on your tail because they can now see the DDD and how much antibiotics you’re consuming… they will then question you, ‘Why are you doing this? What’s the story behind it?’”—IDI03, Human Health “This data is fed back to NARCC and MOH… the main driver for providing feedback is so that you can see your hospital’s performance. This is not meant for comparison, but hospitals are very keen to compare anyway… most of the time the Chairman of the Medical Board take it positively.”—IDI08, Human Health “At one time, [institution] had very bad rates for a certain parameter. Their first step was to check the statistic, case definitions… if it is generally higher than other hospitals, they will go and find out why… The idea of providing feedback helps trigger people because they stand out among the rest…”—IDI08, Human Health “When we started ASP, we proposed for KPIs to be part of [institution] bonus metrics. If you meet it, MOH will know that you are doing good… there will be more funding so that you can reward your people.”—IDI02, Human Health Private hospital “In public hospitals, there’s oversight… if a doctor is prescribing very differently from others, we can feedback to the chair of medical board… But in a private setting there’s no such mechanism.”—IDI12, Human Health Primary care “We conduct reviews and discuss with our polyclinic doctors the gaps in terms of over prescribing or inappropriate prescription of antibiotics for the condition that they see.”—IDI06, Human Health Community “The purpose of surveys is to know the KAP and how to improve the messaging… What other areas we should do?”—IDI16, Human Health
	Environment	“In our list of contaminants of emerging concern that we survey… We work with [organisation] to find out what is their latest list of drugs that are used… Then we look at that at our data… if this is not really used now and there’s no detection over the years, maybe we can drop it and pick something else.”—IDI13, Environment

AMR = antimicrobial resistance; NIR = national immunisation registry; FAO = Food and Agriculture Organization; OIE = Office International des Epizooties; WHO = World Health Organization; AMRCO = antimicrobial resistance coordinating office; DDD = defined daily dose; NARCC = national antimicrobial resistance control committee; ASP = antimicrobial stewardship programme; KPIs = key performance indicators; KAP = knowledge, attitudes, and practice.

### 3.8. Sustainability

The majority of the participants agreed that sustainability, in terms of both financial and non-financial resources, was important to the continuance of NSAP activities. Various priority areas in the implementation domains above were described by the participants, who highlighted operational aspects regarding the assurance that plans are sustainable. One participant explained that the selected priority areas for implementation were largely dependent on available resources, which were considered when developing workplans.

#### Expansion Plans

Expansion plans in the human health sector were highlighted more than other sectors, specifically regarding the inclusion of private hospitals, polyclinics, and GPs for activities such as AMU and HAI surveillance as well as ASPs. A few participants also highlighted novel strategies such as using artificial intelligence to reduce manpower constraints in ASPs.

Animal health participants elaborated on the expansion of AMU regulation. One participant stated that although the goal was to develop legislation and enforce regulations for AMU in the animal health sector, they intended to start with an educational angle to engage relevant stakeholders first.

**Table 9 antibiotics-12-01258-t009:** Subthemes under the theme of sustainability and quotes selected from among them.

Subtheme	Sector	Representative Quotes
Sustainability	One Health	“The NSAP is going to be sustainable. It depends on what is the workplan that we craft out for the next five years. We’ll plan and think about whatever resources we have before we think of the next workplan”—IDI12, Human Health
Expansion plans	Human health	Hospital “We were trying to engage more wards… ICUs are part of the ASP and we have gone on to other major wards. There are still some wards which are blindsided to stewardship… Ob-Gyn, neonatology and paediatrics wards for example…”—IDI03, Human Health “Stewardship is sustainable, but we can’t keep on using manpower so that’s why we need AI to influence prescribers.”—IDI02, Human Health
	Animal health	“…mandatory reporting of AMU… that is an eventual goal. Right now, we are starting small so that we don’t overwhelm them. We start with an educational approach and then reinforce that message… stepping up slowly to something that is more enforcement or mandatory based.”—IDI14, Animal Health

NSAP = national strategic action plan; ICUs = intensive care units; ASP = antimicrobial stewardship programme; AI = artificial intelligence; AMU = antimicrobial use.

## 4. Discussion

Our study assessed the implementation of activities in Singapore’s NSAP using an AMR governance framework. This research included activities carried out prior to the launch of the NSAP in November 2017 up until the end of data collection in October 2021. Overall, we found that as in other countries, the implementation of activities to combat AMR was more advanced in the human health sector compared to other sectors across most areas in the ‘implementation tools’ domain of the AMR governance framework. Some examples included (1) a system for AMR and AMU surveillance in hospitals, (2) hospital AMS services and legislation to optimise AMU, (3) national children and adult vaccination programme for IPC, (4) multiple avenues for education and awareness for both professionals and the public, and (5) extensive research and collaboration networks with many sources of funding. However, there were some areas that were lacking, presenting problems including (1) an incomplete surveillance system for AMR and AMU across all sectors, (2) the need for better AMS and legislation in some sectors, (3) insufficient innovation in education for sustained behavioural modification, and (4) the need for more open research collaborations and the translation of research into policy outcomes.

Although systems for AMR and AMU surveillance have been developed in the hospitals, the participants highlighted that such systems were limited in the primary care and animal health sectors. Most of the AMR surveillance activities in these sectors were performed opportunistically on patients or companion animals as part of a disease workup, while AMU surveillance was difficult due to the large number of GP and animal practice clinics. Other studies reported similar findings of some surveillance efforts in the human and animal health sectors as well as an almost non-existent surveillance system in the environmental sector [18,19,20,21,22]. The reasons mentioned included the fragmented surveillance efforts in the human health sector, difficulty collecting data from private clinics, and insufficient overall technical support across sectors. Our study participants mentioned the development of a national integrated system across various sectors, much like what has been highlighted in Hong Kong and Thailand [18,22]. However, there were concerns that each sector might have their own areas of focus and concern, resulting in data that might not be interpretable and useful across the One Health spectrum. Given that AMR is a One Health issue that affects humans, animals, and the environment, it is important to develop systems that shift away from anthropocentrism, which prioritises human beings as ‘morally superior to everything else in the natural order’, while focusing on a dynamic form of interaction where humans are an equal component in a reciprocal and symbiotic relationship with the rest of nature [23]. An integrated AMR and AMU surveillance system will provide a clearer idea of the country’s AMR situation, possibly elucidating the transmission pathways of certain microorganisms and providing evidence useful for the development of appropriate strategies for optimal AMU [24].

AMS is a crucial component of AMU optimisation. In Singapore, ASPs were introduced in all public acute care hospitals since 2011 [10]. The MOH provided SGD 20 million in funding for the development of these ASPs in terms of manpower and infrastructure support. Hospital ASPs have been described to be successful, as the participants mentioned the improvement in intervention acceptance and appropriate prescribing rates over the years. However, there were areas for improvement. AMS in private hospitals, primary care, and the animal health sector were described as rudimentary or non-existent. For the latter two sectors, a chapter on how to set up an ASP in veterinary practice has been included in the Prudent Use of Antimicrobials Guidelines, but relevant guidelines for AMU in the primary care setting have been described to be lacking and outdated [17]. Regionally, the implementation of ASPs in the hospital and primary care setting has been mentioned in Hong Kong and the Philippines, but not with respect to the animal health or environment sectors [18,21,25,26]. The AMS situation at the global level was similar: it was more established in the human health sector than in the animal health sector, the latter of which has been gaining in prominence in recent years [27].

Other than AMS, regulations also help optimise AMU. Regarding the human health sector, access to antimicrobials was restricted because these drugs were classified as prescription-only medicines; that is, they were only accessible with a valid prescription from a physician. In addition, the unauthorised online sale of prescription medicines has been monitored closely by the HSA, which screened and stopped illegal shipments of medicine entering Singapore [28]. Concerning the animal health sector, the former AVA had in place licensing conditions for the manufacture of animal feed and implemented directives to explicitly prohibit the use of antibiotics for growth promotion in livestock and aquaculture. A study of Singaporean food-fish farmers conducted to understand the patterns and determinants of antimicrobial use showed that local regulatory factors constituted one of the reasons behind the absence of AMU for growth promotion and infection prevention in addition to individual and market factors such as personal knowledge regarding AMR and the narrow profit margin associated with AMU, respectively [29]. Some participants believed that other regulations on veterinary medicines were still rather lax, remarking that pet owners can purchase antimicrobials without a valid veterinary prescription as there is no legislation to restrict antimicrobial sales in this field. This is an area that agencies are working on in order to establish a framework for veterinary drug registration and for veterinary prescription for drugs used in all animal sectors [30]. Studies conducted in Tanzania and the Philippines have shown that there are similar medicinal regulations in place to ensure appropriate AMU. However, there were problems in implementation given the lack of capacity and resources [20,21]. Antimicrobials for human or animal use can easily be purchased over the counter in such settings [19,21]. It is important to note that the ‘hard’ approach through legislation and regulation may not always work. For example, Thailand attempted to reclassify several antibiotics as prescription-only drugs in 2016, but this aroused disagreements among health professionals, objections from pharmaceutical companies and retailers who feared reduced profits from sales, and patients’ concerns about the possible reduced access to antibiotics or the need for additional fees to obtain them [31]. This is similar to the discussion in 2005 on the issue of separating drug prescribing rights from dispensing rights in Singapore, which faced backlash from both the medical community and the patients [32]. Therefore, when incorporating legislation and regulatory tools into NAPs, it is important to engage all relevant stakeholders and consider their roles throughout the implementation process.

Very often, a mix of ‘hard’ and ‘soft’ approaches is required for policy implementation as a wide range of factors including trust, framing, and perceived effectiveness, spanning individual and contextual factors, affect preferences to the approaches [33]. Our participants stated that even with AMU guidelines and legislation to regulate AMU, the issues of improper diagnoses and the lack of regulation of physician prescriptions were still present. The latter was shared by a participant as a case of home-administered intravenous meropenem, an antimicrobial usually given to hospitalised patients with serious infections, carried out by a GP prior to hospitalisation. This problem was compounded by patients’ erroneous beliefs about the role of antimicrobials in the treatment of their conditions, which resulted in the expectation of receiving antimicrobials from their respective physicians [34]. This highlights a need for better education and communication, constituting one of the ‘soft’ approaches that have been discussed by our participants at great lengths.

Public awareness campaigns on AMR have been highlighted in other countries in various forms including social media and physical events [18]. The campaigns highlighted often tied in with the annual WAAW celebrations [18,19,20,21,35]. In addition, education for professionals was also discussed to various degrees, and across the countries, there seemed to be more educational efforts exerted toward the human health sector than the animal health sector [19,20,21]. The situation in Singapore was similar, as there were more efforts exerted toward human health than animal health in terms of both professional education and public awareness. There were concerns about the effectiveness of these educational efforts in promoting behavioural changes given that most of them were short-termed and only discussed superficially with the same mode of delivery. A recent survey conducted on 2004 respondents showed that there was still poor knowledge regarding AMR and AMU; for example, only 9% of the respondents understood that AMR occurs in bacteria and not in the human body [36]. To facilitate potential behavioural modifications, innovation in new educational methods is required for the better delivery of messages, with consideration given to why people behave in the way they do with regard to antimicrobials. Some of the methods suggested include (1) the ‘five stages of change model’, including pre-contemplation, contemplation, preparation, action, and maintenance; (2) antimicrobial mainstreaming to make it an integral part of all policy planning and implementation; and (3) the provision of social information on physicians’ antimicrobial prescription and patient’s antimicrobial intake [37,38].

Research on AMR can help generate evidence for translation into health policies and the improvement of strategies to tackle AMR. In Singapore, AMR research collaborations have been prioritised, as seen in the establishment of OHARP. In addition, participants also described multiple channels of funding and AMR research topics that were conducted, ranging from bench-top research to KAP studies, with more research on the animal health and environmental sectors being conducted recently. However, there were also some issues highlighted by the participants, including a lack of truly open collaboration with constraints on data sharing, less AMR research outside the human health setting, and a lack of translation from research to policy outcomes. A recent study conducted to understand the challenges and opportunities of the AMR research collaboration landscape in Asia presented similar findings, including a difficulty establishing collaboration due to a preference for working with like-minded professionals from one’s own field and an impediment to trust due to poor communication and commitment [39]. Given that AMR is a One Health issue, further actions can be taken to develop new AMR research networks and improve existing networks between stakeholders both across and within sectors. In addition, to improve trust, collaborations should develop transparent and flexible agendas with shared leadership, research priorities, and the usage of secure data-sharing platforms.

Table 10 summarises the policy recommendations identified from the study results, participant recommendations, and the available literature [21,22,24,30,37,38,39,40,41].

To our knowledge, this is one of the first qualitative studies investigating the implementation of activities in the NSAP of Singapore. This study complements our other paper, which described the policy process and development of the NSAP using the same methodology and participant population. We were able to analyse our data in a structured manner, following the AMR governance framework that we had developed previously. The participants recruited included a range of stakeholders in the human health, animal health, and environmental sectors. Unfortunately, there was a greater level of representation from the human health sector compared to that from the animal health and environment sectors. This could be because the human health sector was much more involved in the AMR-related activities and because stakeholders in general were busy with the pandemic and unable to allocate time to participate in our study. Pandemic restrictions also affected our data collection process as we could only conduct interviews virtually, which could have limited rapport building. Despite these limitations, we believe that we reached thematic saturation, as no new themes emerged from the data, and the existing themes were rich in content. In addition, our study is qualitative, following an interpretative approach that focuses on the participants’ perspectives. As such, there may be a few inaccuracies with respect to some of the national programmes and structures. We have tried our best to minimise such inaccuracies via the member check process, wherein participants review a draft manuscript and make suggestions where appropriate, and to explicitly mention that the description of the programmes and structures was based on the participants’ perceptions. Finally, while we have attempted to understand the implementation of activities in the NSAP, we did not intend to quantify or evaluate the success of the NSAP’s implementation. This is because some activities, especially those in the human health sector, were implemented before the launch of the NSAP. At the time of writing, there are plans to embark on such an evaluation to better assess the NSAP’s implementation. This could involve both qualitative and quantitative evaluations, executed using selected indicators, on the progress of the NSAP’s implementation, including the effectiveness and efficacy of its programmes in achieving its stated objectives. This will help identify further areas for improvement and provide recommendations for future strategies.

## 5. Conclusions

We examined the implementation processes of activities in the NSAP of Singapore using an AMR governance framework and found that it was much better implemented in the human health sector than in the other sectors across most areas of the framework. There is a need for better surveillance and AMS and AMU regulations, especially outside the public hospital setting. In addition, new methods of education and the delivery of messages should be considered for sustained behavioural modifications towards AMR along with stronger support for AMR research networks. Improvements in these areas will enhance the overall implementation of the NSAP through a more holistic One Health approach.

## Figures and Tables

**Figure 1 antibiotics-12-01258-f001:**
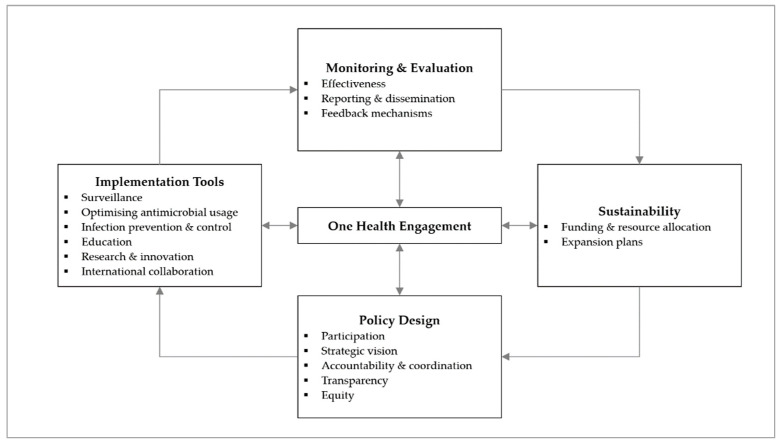
Governance framework for the assessment of national action plans on antimicrobial resistance (adapted from Chua et al., 2021) [12]. Reuse is licensed under an open access Creative Commons CC BY 4.0 license.

**Table 1 antibiotics-12-01258-t001:** Summary of participants with respect to type of institution and sector.

Type of Institution	Sector			Total
	Human Health	Animal Health	Environment	
Academia		1	1	2
Government agency	6	3	2	11
Hospital (public and private)	5			5
Industry	1			1
Primary care	1			1
Total	13	4	3	20

**Table 10 antibiotics-12-01258-t010:** Policy recommendations.

S/N	Policy Recommendations	Details
1	Develop integrated surveillance systems for AMR and AMU across all sectors	The current AMR and AMU surveillance systems are largely limited to the hospital setting and should be extended to the primary care and animal health sectors. Surveillance systems should be integrated across sectors to provide a more holistic and complete AMR and AMU situation in the country.
2	Extend AMS services to all relevant sectors	The current AMS services are largely limited to the public hospital setting and should be extended to the private hospital, primary care, and animal health sectors.
3	Strengthen AMU regulations, especially in the animal health sector	Enhance regulations for AMU in veterinary medicine, for example, through veterinary drug registration and the legislation of mandatory veterinary prescriptions for drugs. Regulations should be developed synergistically with all stakeholders throughout the process to ensure their buy-in. Adequate monitoring and enforcement programmes should be in place to ensure successful implementation of the regulations.
4	Develop novel educational strategies for sustained behavioural modification	Innovation in new educational methods is required for better delivery of messages to a targeted audience. The reasons behind individuals’ behaviours regarding antimicrobials should be considered.
5	Improve physician diagnoses through education	Education frameworks that include a knowledge base as well as perspectives and attitudes about diagnosis are important. More accurate diagnoses reduce the risk of inappropriate AMU.
6	Enhance support for AMR research networks	Provide more funding to support cross-sectoral AMR research with flexible agendas as well as shared leadership and research priorities. Develop efficient data-sharing systems to improve transparency, which, in turn, may improve trust and enhance collaboration.
7	Improve the translation of research into policy and action	Aspects regarding implementation science and community engagement including research and policy development processes should be considered in order to maximise the policy impacts of research and improve population outcomes. This consideration should be accompanied by appropriate monitoring and evaluation using published indicators as well as grassroot indicators that are relevant to the stakeholders.
8	Strengthen partnerships at the international and regional levels	Collaboration and network building provides an avenue for better resource management and the capacity to execute the implementation plans, providing an opportunity for knowledge generation and innovation. Initiatives such as policy dialogues and workshops between policymakers and researchers could enhance collaboration, allowing individuals to learn from each other’s experiences and providing funding opportunities.

AMR = antimicrobial resistance; AMU = antimicrobial use; AMS = antimicrobial stewardship.

## Data Availability

Not applicable.

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
