# Peer review of "A Qualitative Study on the Implementation of the National Action Plan on Antimicrobial Resistance in Singapore"

_antibiotics, 2023, doi:10.3390/antibiotics12081258_

Round 1

Reviewer 1 Report

Your article is outstanding; I found tables presenting selected quotes of great value for other countries' readers, and finally, I found Table 10, Policy recommendations, a great start for countries trying to implement action plans like yours. 

Author Response

Many thanks for your kind comment as well as the time taken to review our article, we really appreciate it. We are also glad that you found the quotes and policy recommendations valuable! 

Reviewer 2 Report

I find this manuscript of interest to a broad audience as antibiotic resistance is an emerging topic in public health. The Implementation of the National Action Plan on Antimicrobial Resistance in Singapore is an important step for AMR control in this part of the world and I find it an excellent template for other countries where this plan is not implemented yet.

The tables, figures, and literature are more than appropriate. I believe this article adds to the body of literature in microbial resistance field.

Author Response

Many thanks for your kind comment as well as the time taken to review our article, we really appreciate it. We hope that this article will generate more interest in AMR-related work globally.

Reviewer 3 Report

Thank you for the opportunity to review this outstanding manuscript describing the state of preparedness with respect to antimicrobial surveillance and use in Singapore. I agree with your study design, structuring the interviews to discuss five areas: human health, primary care, animal health, food, and water (environmental). The results clearly point to areas for improvement, and highlight the (not surprising) more advanced framework present in hospitals. Your discussion thoroughly discusses and explains these trends, and includes a well organized set of policy recommendations to address identified deficiencies. This manuscript was a pleasure to read.

Author Response

Many thanks for your kind comment as well as the time taken to review our article, we really appreciate it. We hope that this article will spark interest in AMR-related work across the One Health spectrum, especially in areas where it is currently not as advanced.